# Reliability and Convergent Validity of Endurance Indices Derived from Near-Infrared Spectroscopy and Electromyography during a Bilateral Hanging Task in Amateur Rock Climbers

**DOI:** 10.3390/jfmk9030161

**Published:** 2024-09-10

**Authors:** Wai-Hang Kwong, Jia-Qi Li, Chun-Hung Lui, Hiu-Tung Luk, King-Fung Lau, Ray Seaby, Ananda Sidarta

**Affiliations:** 1Department of Rehabilitation Sciences, The Hong Kong Polytechnic University, Hong Kong, China; jiaqiqi.li@connect.polyu.hk (J.-Q.L.); chun-hung-jack.lui@connect.polyu.hk (C.-H.L.); hiutung.luk@connect.polyu.hk (H.-T.L.); kingfung.lau@connect.polyu.hk (K.-F.L.); 2Shenzhen Institutes of Advanced Technology, Chinese Academy of Sciences, Shenzhen 518000, China; 3Curtin School of Allied Health, Faculty of Health Sciences, Curtin University, Perth, WA 6102, Australia; ray.seaby@curtin.edu.au; 4Rehabilitation Research Institute of Singapore, Nanyang Technological University, Singapore 308232, Singapore; ananda.sidarta@ntu.edu.sg

**Keywords:** electromyography, endurance, near-infrared spectroscopy, rock climbing, amateur

## Abstract

**Background:** The ability to hang for a long time before forearm muscle fatigue is a crucial element of successful rock climbing. Electromyography (EMG) and near-infrared spectroscopy (NIRS) are also useful for measuring hemoglobin oxygenation for determining muscle endurance. In the present study, we aimed to evaluate the reliability and validity of muscle endurance indices derived using EMG and NIRS during a hanging task. **Methods**: A bilateral hanging task was designed to compare rock climbers and non-climbers in terms of the slopes of changes in the median frequency (MDF) and tissue oxygenation index (TOI) of forearm muscles. **Results**: A total of 17 participants were included in each of the two groups. The intraclass correlation coefficient (3,1) values derived for the MDF slope, TOI slope, ΔTOI, percentage change in oxygen concentration, and ΔHbt were 0.85, 0.73, 0.65, 0.75, and 0.65, respectively. The MDF slope, TOI slope, and ΔHbt differed significantly between the groups (*p* < 0.05). The MDF slope, TOI slopes, and ΔHbt were significantly correlated with V-scale levels for climbing (*p* < 0.05). **Conclusions:** The satisfactory reliability and observed distinctions between climbers and non-climbers imply that these indices are a valuable tool for assessing muscle endurance.

## 1. Introduction

Rock climbing is a popular sport and has recently been recognized as a competitive activity. In the 2021 Summer Olympics, sport climbing—a form of rock climbing—was officially debuted as a medal sport. Physiologically, rock climbing is characterized by several repetitions of isometric contractions. Through sustained and intermittent contractions, the upper extremities support anchoring [1], and strength in the flexor muscles of the fingers may be a good indicator of climbing ability [2]. By contrast, the lower extremities support balancing, anchoring, and the upward propulsion of the body [3]. Therefore, rock climbing requires muscle strength and endurance, in addition to the coordination between the upper and lower extremities.

A study reported that the time to fatigue during hanging among rock climbers was twice as long as that among non-climbers, as revealed by a decreased median frequency (MDF) on electromyography (EMG) scans [4]. A longer time to fatigue of upper body muscles may increase ‘hanging time’, which is deemed crucial for successful rock climbing.

Studies have used EMG to investigate muscle activation in rock climbers and to quantify time to fatigue, which can be considered a proxy measure for muscle endurance [4,5,6]. For example, a study applied EMG to examine endurance at 80% maximal voluntary contraction force of extrinsic hand flexor muscles—flexor digitorum superficialis (FDS) and flexor digitorum profundus (FDP)—and also the extensor muscle—extensor digitorum communis (EDC)—using a dynamometer fingertip force task mimicking a rock climbing grip; the results revealed that the endurance of elite climbers was twice that of non-climbers [4].

EMG does not fully disclose physiological responses underlying better muscle endurance performance. Near-infrared spectroscopy (NIRS) is a noninvasive technology and has recently been used to evaluate muscle endurance by continually monitoring regional tissue oxygenation [7,8,9,10,11]. NIRS can provide various measurements that can be used to evaluate muscle endurance on the basis of hemodynamics [12]. The tissue saturation index (TSI) is commonly used to measure the concentrations of oxygenated and deoxygenated hemoglobin, respectively [7,8,9]. The half-time of oxygen consumption recovery (O_2_HTR) has also been used to assess the oxidative capacity index of the gastrocnemius and soleus muscles after an ankle plantar flexion exercise [13]. A study reported a correlation between O_2_HTR and muscle endurance during forearm muscle activation [14]. Muscle oxygen consumption is another performance index [15] that can be used to evaluate tissue oxygenation capacity by calculating the rate of change in the concentration of deoxygenated hemoglobin. Moreover, muscle oxygenation can be evaluated by measuring the concentrations of oxygenated hemoglobin and myoglobin by varying the NIRS wavelengths [16]. For example, Baláš et.al. revealed that muscle oxygen consumption in isolated finger flexion and exhaustive incremental treadmill tests was correlated with muscle endurance during climbing [17].

As of our current understanding, the literature lacks sufficient evidence on the reliability of various NIRS measures for assessing forearm muscle endurance in climbing-related activities. Additionally, it remains unclear whether participation in climbing as a recreational activity can enhance muscle endurance. This study aims to evaluate the reliability of endurance indices derived from both EMG and NIRS during the hanging task. Additionally, we will investigate the discriminant validity of these indices by comparing climbers and non-climbers. Furthermore, the study will explore the construct validity by assessing the correlation of these indices with climbing ability within the climber group.

We hypothesize that the endurance indices derived from both EMG and NIRS during a hanging task will demonstrate good test–retest reliability. The hanging task is particularly relevant in the study because it closely simulates the demands placed on the forearm muscles during actual rock climbing. This task is used because it mimics the isometric contractions and sustained grip required in climbing, providing a more valid assessment of forearm endurance in climbers. Furthermore, we hypothesize that climbers will demonstrate greater endurance than non-climbers, due to engagement in climbing strengthening these muscles. It is anticipated that these indices will exhibit discriminant validity by revealing significant differences between climbers and non-climbers. Additionally, the study posits that the indices will show construct validity by correlating significantly with climbing ability within the climber group.

## 2. Materials and Methods

### 2.1. Participants

#### 2.1.1. Inclusion and Exclusion Criteria for Amateur Rock Climbers

Amateur rock climbers were recruited from a local climbing gym through convenience sampling, including word of mouth and snowball sampling. Amateur climbers were defined as those who did not participate in climbing competitions at a professional level or engage in structured training programs aimed at becoming professional climbers. The eligibility criteria were as follows: being aged between 18 and 40 years, participating in rock climbing activities at least twice a week (regular climbers), and having the ability to follow instructions and provide informed consent. Exclusion criteria encompassed individuals with a history of upper extremity pain or injury within the past month, those who had undergone upper extremity surgery, individuals taking medications causing vessel dilation or constriction, those with excessive caffeine intake, or those with any other medical conditions that could impede the assessment.

#### 2.1.2. Inclusion and Exclusion Criteria for Non-Climbers

Non-climbers were recruited through convenience and snowball sampling methods using a poster advertisement on a university campus. The eligibility criteria for non-climbers were as follows: being aged between 18 and 40 years and having the ability to follow instructions and provide informed consent. The exclusion criteria were similar to those used for the climbers.

#### 2.1.3. Sample Size Estimation

We prospectively estimated the study sample size through G*power (v3.1.0; Franz Faul, University of Kiel, Germany) with an independent sample *t*-test, with the α level being set to 0.05 and power being set to 0.8. The estimation was based on the study conducted by Vigouroux et al. [4], which revealed a large difference (Cohen’s d = 3.8) between trained climbers and non-climbers in terms of the MDF of the forearm flexor after a repetitive gripping test. We selected a more conservative effect size of 1.0 for our estimation. Thus, the total sample size was 34; each group comprised a total of 17 participants.

### 2.2. Assessment Procedure

Before the experiment, the research personnel verbally explained the study procedure to eligible individuals. All participants signed a consent form approved by the institutional review board (HSEARS20211111002). This study was conducted at the rehabilitation laboratory of Hong Kong Polytechnic University. The participants were invited to participate in two assessment sessions, and a 1-week interval was ensured between the two sessions. Each assessment session lasted approximately 1 h.

#### Bilateral Hanging Task

All participants performed the bilateral hanging task three times by using a horizontal bar. The hanging time was set to be a minimum of 20 s on the basis of the results of a feasibility trial conducted by the research team or until the participant decided to stop. All non-climbers who participated in the trial reported that they could endure the hanging task without any adverse effects. The participants rested for 5 min between trials. The participants performed stretching and warm-up exercises before the test. Practice trials were conducted to ensure the participants’ familiarity with the task.

### 2.3. Outcome Measures

#### 2.3.1. Demographics

Participant demographics, including age, sex, body height, body weight, and dominant hand, were recorded. In addition, the frequency of rock climbing, duration of participation, and level of climbing were recorded. The level of climbing was measuring using the V-scale, starting at V0 and going up to V17. The higher the number in the V-scale, the greater the difficulty of a bouldering climbing problem [18].

#### 2.3.2. EMG

The Delsys Trigno Avanti Research+ wireless surface EMG acquisition system (Delsys Inc., Natick, MA, USA) was used to collect surface EMG signals at 4000 Hz. The methods used for skin preparation and EMG electrode placement primarily followed those described by Hermens et al. [19]. The skin was prepared by shaving the skin surfaces where the sensors were placed and cleaning the areas using alcohol swabs. The EMG electrodes were placed on the wrist flexor [20]. Kim suggested the that FDS and FDP are used more than the other flexors in gripping tasks [21]. Vigouroux et al. recorded the EMG signals from FDP, FDS, and EDC during an intermittent exercise to compare the difference between elite climbers and sedentary individuals [4]. Guo F. et al. showed that the FDS EMG signal was the highest, followed by the biceps brachii (BB) and latissimus dorsi during a 15 m speed climbing [22].

#### 2.3.3. Muscle Oxygenation

Muscle oxygenation was assessed using a wireless artinis PortaMon NIRS system (Artinis Mecical System, Einsteinweg, The Netherlands) at 20 Hz. Functional NIRS is a noninvasive method of evaluating the hemodynamic responses of various tissues, such as the cortex and the muscle. The optical absorption features of the NIRS spectra (600–900 nm) differ between oxyhemoglobin and deoxyhemoglobin [23]. Therefore, real-time changes in muscle oxygenation during exertion can be observed using NIRS. In this study, NIRS sensors were placed over the FDS and FDP of the dominant hand (Figure 1). Table 1 summarizes the metrics of the near-infrared spectroscopy being used in different studies.

**Figure 1 jfmk-09-00161-f001:**
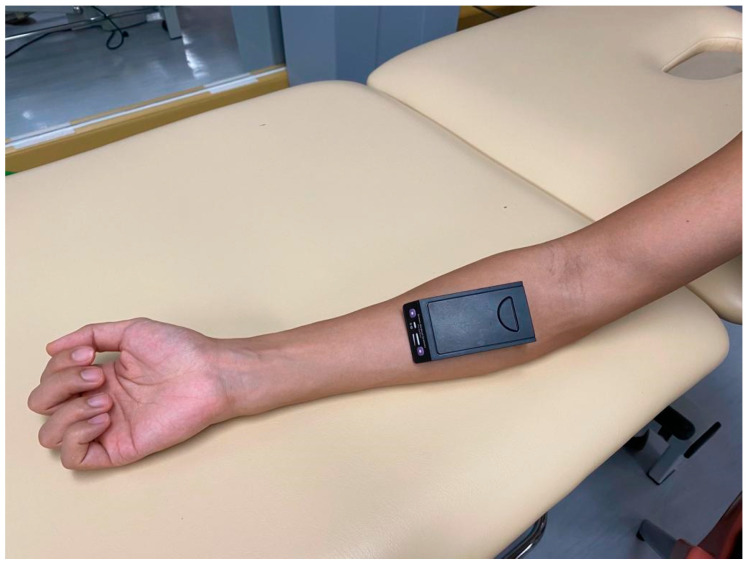
Near-infrared spectroscopy sensor placement location.

**Table 1 jfmk-09-00161-t001:** Endurance indices assessed by near-infrared spectroscopy.

Study	Metric	Parameter Measured	Indication
Taelman et al. [7]	Change in tissue oxygenation index (ΔTOI) during muscle contraction	ΔTOI TOI slope	ΔTOI negatively correlates with the total exertion time, suggesting higher deoxygenation results in early exhaustion (Figure 2A).
Muramatsu et al. [8]	Difference in oxyhemoglobin and deoxyhemoglobin (ΔHbt) after the exertion	ΔHbt	ΔHbt increases according to the elapsed time in case of exhaustion (Figure 2B).
Ferguson et al. [24]	% change in oxyhemoglobin	Oxyhemoglobin	The large % change in oxygenated hemoglobin resulted in only a small % change in muscle saturation, revealing how the energy metabolism of the system attempts to resist fatigue (Figure 2B).

**Figure 2 jfmk-09-00161-f002:**
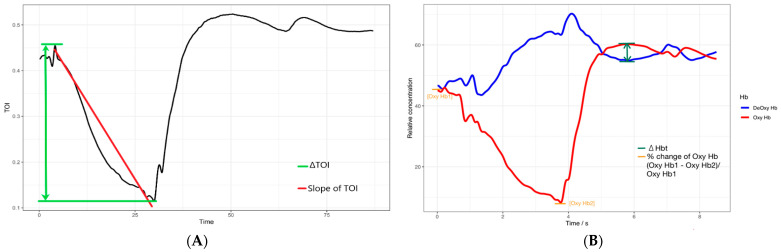
Illustration of the NIRS’s endurance indices. (**A**) The red line represents the slope of TOI changes during the hanging task, and the green lines indicate the TOI changes before and after the hanging task. TOI, tissue oxygenation index. (**B**) The green line indicates the difference between the concentrations of oxyhemoglobin (Oxy Hb) and deoxyhemoglobin (DeOxy Hb) after the hanging task, and the orange lines indicate the percentage changes in Oxy Hb concentration before and after the hanging task.

### 2.4. Data Processing

EMG data were processed using EMGworks (ver.4.7.9; Delsys Inc.). The data were processed in accordance with the recommendations of the International Society of Electrophysiology and Kinesiology for surface EMG [25]. The data were digitally filtered using a Butterworth filter with a high-pass frequency of 5 Hz and a low-pass frequency of 500 Hz.

A Fourier transform was performed with a time epoch of 250 ms to obtain the MDF of each muscle during the hanging task. Moreover, TOI was calculated (1) to estimate the dynamic balance between oxygen supply and consumption in tissues [7]:(1)$$TOI=\frac{k\;\cdot \;{HbO}_{2}}{k\;\cdot \;{HbO}_{2}+k\;\cdot \;HbR}$$

*HbO*_2_ is the concentration of oxyhemoglobin, and *HbR* is the concentration of deoxyhemoglobin. The constant “k” represents a proportionality factor used to scale the concentrations of HbO and deoxygenated HHb. Although “k” cancels out in the formula, it was included in the formula to acknowledge the general form of such equations.

### 2.5. Data Analysis

Participant demographics were analyzed using descriptive statistics. The mean values of the indices during the three repeats were determined for statistical analysis. Gradual changes in MDF were modeled using linear regression. Between-group differences in the indices were determined using an independent *t*-test and Mann–Whitney U tests. We assessed normality using the Shapiro–Wilk test and confirmed it with Q-Q plots. The homogeneity of variances was checked with Levene’s test. Bland–Altman analyses and intraclass correlation coefficient (ICC) values were used to measure the test–retest reliability of the indices in both groups. ICC(3,1) values were calculated to investigate the consistency of measurements between the two trial sessions; the measurements made during each session were averaged for analysis. Spearman correlation coefficient analysis was conducted to evaluate the statistical correlations between indices for measuring forearm muscle endurance. The ICC was classified according to the guidelines by Koo and Li [26], with values below 0.5 indicating poor reliability, 0.5–0.75 indicating moderate reliability, 0.75–0.9 indicating good reliability, and above 0.9 indicating excellent reliability. Cohen’s d effect sizes were interpreted as small (0.2), medium (0.5), and large (0.8), following Cohen’s criteria [27]. A *p*-value of less than 0.05 was considered statistically significant for all analyses. All data analyses were conducted using R (v4.1.2; R Core Team).

## 3. Results

A total of 17 climbers and 17 non-climbers were recruited between September 2021 and February 2022. Table 2 presents a summary of the demographics of the participants. No statistically significant differences were noted between the climbers and non-climbers in terms of age, sex, or body mass index (*p* > 0.05).

The ICC analysis revealed good to moderate reliability. The ICC(3,1) value for the MDF slope was 0.85, suggesting good reliability (0.75 < ICC < 0.9); by contrast, moderate reliability was noted for the TOI slope, change in TOI (ΔTOI), percentage change in oxygen concentration, and difference between oxyhemoglobin and deoxyhemoglobin concentrations (ΔHbt; 0.5 < ICC < 0.75) [26]. Table 3 presents the test–retest reliabilities of the outcome measures.

Bland–Altman plots were used to evaluate the agreement between the two assessments in five different outcome measures (Appendix A). In our Bland–Altman analysis, we calculated the mean difference, standard deviation, and upper and lower limits of agreement. In general, no bias was detected in the outcome measures because the bias was extremely close to 0 for all parameters.

Table 4 presents the means and standard deviations of the MDF slope, TOI slope, ΔTOI, percentage change in oxygen concentration, and ΔHbt measured in both assessments for all participants, the climbers and the non-climbers. The MDF slope, TOI slope, and ΔHbt (*p* < 0.05) were found to be valid for discriminating between the two groups, with the corresponding effect sizes being 0.93, 0.91, and 0.93, respectively.

Regarding the correlation between the V-scale and indices in the climbers (Figure 3, Figure 4 and Figure 5, Table 5), the MDF slope, TOI slope, and ΔHbt demonstrated significant correlation with the V-scale (*p* < 0.05).

## 4. Discussion

In this study, we assessed the reliability of different endurance indices obtained through EMG and NIRS. The findings indicate that the MDF slope demonstrated good test–retest reliability, whereas endurance indices derived from NIRS exhibited moderate reliability. Additionally, the MDF slope, TOI slope, and ΔHbt effectively differentiated between the climber and non-climber groups, showing an association with climbing ability.

Regarding the reliability of the MDF and various hemodynamic indicators for measuring muscle endurance, the MDF slope exhibited good reliability. A study suggested that at least 30 heterogeneous participants should be included to effectively evaluate the reliability of measurements [28]; the present study included a total of 34 participants with different levels of climbing experience. The MDF slope appeared to evaluate muscle endurance. The hemodynamic indicators, including the TOI slope, ΔTOI, and percentage change in oxyhemoglobin concentration, exhibited moderate reliability levels. However, studies using ICC analysis to assess the reliability of hemodynamic indicators measured using NIRS have reported that the indicators exhibited good to excellent reliability levels for isometric or isotonic muscle contractions [29,30,31]. A possible explanation for this difference is that the time required for the hanging task in the present study was 20 s, which is longer than the time required for isometric/isotonic muscle tasks in the aforementioned studies. In addition, the tasks in the aforementioned studies were conducted at a specific percentage of maximum voluntary contraction (MVC); by contrast, the hanging task could not be conducted at a specific percentage of MVC. This implies that rock climbers and non-climbers performed a hanging task at different percentages of MVC because of the differences in training received.

In the surface EMG power spectra, a shift in MDF toward a low frequency during sustained muscle contractions is a manifestation of localized muscle fatigue [32,33]. In the present study, the MDF slopes exhibited a downward shift in both groups. A decrease in MDF during sustained isometric contractions is associated with a reduction in muscle fiber conduction velocity (MFCV) [34,35]. We noted a significant difference between the climbers and non-climbers in terms of MDF slope, suggesting that MDF decreased at different rates between the two groups. A smaller reduction in MDF was observed in the climbers, signifying a smaller reduction in MFCV. Owing to the nature of rock climbing, the intramuscular pressure during sustained muscle contractions is higher than that during dynamic contractions, which prevents blood flow and engenders the accumulation of metabolites, such as lactic acid; this increases the intramuscular pH, which reduces the MFCV [36]. A relatively low rate of MDF reduction indicates that rock climbers develop high tolerance to accumulated metabolites [37].

TOI represents the dynamic balance between oxygen supply and consumption in a target muscle group [38]. In this study, the rate of change and between-group differences in TOI were measured during the hanging task to observe the deoxygenation patterns of the participants [39,40,41]. The two groups could be identified according to the TOI slope. The TOI slope observed for the climbers was less negative than that observed for the non-climbers. This implies that the non-climbers exhibited a higher rate of deoxygenation than the climbers, suggesting that non-climbers feel tired more rapidly during a hanging task than rock climbers. A previous study demonstrated that participants with a less negative TOI slope maintained the exertion task longer than those with a faster drop in TOI [7]. These findings suggest that in addition to having an excellent ability to maintain oxygen supply to working muscles, rock climbers exhibit an improved oxygen utilization capacity [42].

A study demonstrated that participants with higher ΔTOI values performed an exercise for a shorter duration and felt tired more rapidly than those with lower ΔTOI values [7]. By contrast, the present study revealed no correlation between grouping and ΔTOI. This might be because the 20 s hanging time was insufficient to cause a considerable difference in muscle endurance between the climbers and non-climbers.

For the climbers, we analyzed V-scale levels to investigate their correlation with the five outcome measures considered in the study. The MDF, ΔHbt, and TOI slopes were correlated with V-scale levels (*p* < 0.05). However, climbers with a V-scale level of 6 exhibited lower values of the aforementioned outcome measures than those with a V-scale level of 5; this result might have been influenced by various factors, including the climbing skills and body conditions of the participants (e.g., flexibility and body coordination). Rock climbing considerably increases lactate levels in the blood, which indicates the anaerobic nature of the sport [1,43,44]. Increased levels of lactate in the blood are associated with decreased handgrip endurance and strength [45,46,47]. Therefore, after prolonged training, rock climbers may better tolerate and alleviate the increased levels of lactic acid. Furthermore, rock climbing engenders a disproportionate increase in heart rate for a given value of the maximum level of oxygen used by the body during this activity (VO_2_) [48,49,50]. This suggests that rock climbing, which requires intermittent isometric contractions of forearm muscles, leads to high increases in heart rate and blood pressure; this may be attributed to the distribution of a high cardiac output to the working forearm muscles without a considerable increase in oxygen consumption. Metabolites accumulated in the working muscles send signals to the central nervous system, an action called metaboreflex [51,52]. Metaboreflex elicits a sympathetically mediated pressor response, thus increasing the heart rate and systemic arterial pressure without affecting VO_2_ [51,52]. Hence, reduced levels of blood pressure in trained rock climbers suggest afferent nerve desensitization and low metabolite production, which, in turn, indicate low muscle metaboreflex activation [1,37,53].

ΔHbt increases as the level of exhaustion increases [8]. During isometric hold tasks, climbers exhibit a faster recovery of oxyhemoglobin in flexor muscles than non-climbers [12]. Thus, the ΔHbt is valid for discriminating between the two groups. A larger ΔHbt value implies a faster recovery of oxyhemoglobin, which is essential for isometric muscle contractions. Anaerobic pathways play a predominant role during the first 10–15 s of muscle contraction; through such pathways, adenosine triphosphate is released from phosphocreatine (PCr) and serves as an energy source for muscle contraction [54]. The resynthesis of PCr is dependent on aerobic respiration [54]; hence, the rate of oxyhemoglobin regeneration governs that of PCr resynthesis. In this study, the climbers exhibited a larger ΔHbt value than the non-climbers, suggesting that the rate of oxyhemoglobin regeneration was faster in the climbers than in the non-climbers. This rate is crucial for a sustained hold during bouldering for a higher altitude. In trained athletes, higher ΔHbt values may be associated with higher levels of oxygen dissociation from hemoglobin [55]; this results in PCr resynthesis and oxyhemoglobin regeneration at rates faster than those noted in non-climbers.

Previous studies demonstrated that elite and advanced rock climbers exhibited higher levels of deoxygenation than healthy non-climbers and reported no differences between intermediate climbers and non-climbers [56,57]. In our study, the climbers were at the intermediate level. Both non-climbers and intermediate climbers may have less extensive capillary bed development in the flexor muscle and limited capacity of oxygen extraction from the perfused blood [58]. Future studies may prolong the duration of the hanging task to effectively differentiate between non-climbers and intermediate climbers.

The present study has some limitations. The indices measuring the physiological aspect of muscle endurance may be subject to various environmental factors (e.g., room temperature and time of the day) and participant factors (e.g., food intake, exercise, and sleep quality) [59,60,61,62]. These factors must be controlled in future studies to obtain more reliable and consistent results. Regarding the V-scale levels of the climbers, the skill levels ranged from beginner to intermediate. This gradient was not sufficiently wide for exploring the chronic adaptation in the muscle physiology of advanced rock climbers through the use of endurance indices. The ICC values derived for ΔTOI and ΔHbt were 0.653 and 0.647, respectively, suggesting moderate reliability.

## 5. Conclusions

This study revealed that the MDF slope exhibited good reliability; by contrast, the TOI slope, ΔTOI, percentage change in oxygen concentration, and ΔHbt exhibited moderate reliability levels. The MDF slope, TOI slope, and ΔHbt are valid for discriminating the climbers and non-climbers. The MDF and TOI slopes were significantly correlated with V-scale levels for climbing. The reliable and valid endurance indices derived from EMG and NIRS during a hanging task offer a means of assessing forearm muscle endurance. The identified differences between climbers and non-climbers, along with the correlations between the indices and climbing ability, suggest practical applications for optimizing training regimens. By including participants who engage in rock climbing recreationally, we aim to demonstrate that significant differences in muscle endurance can emerge, even among those with relatively little experience, emphasizing the impact of regular climbing activities on muscular adaptation.

## Figures and Tables

**Figure 3 jfmk-09-00161-f003:**
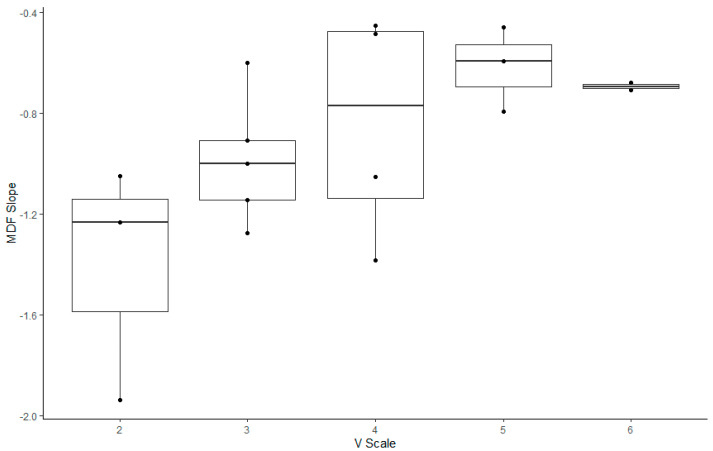
Box plot illustrating the slope of the changes in the median frequency of rock climbers with various V-scale levels.

**Figure 4 jfmk-09-00161-f004:**
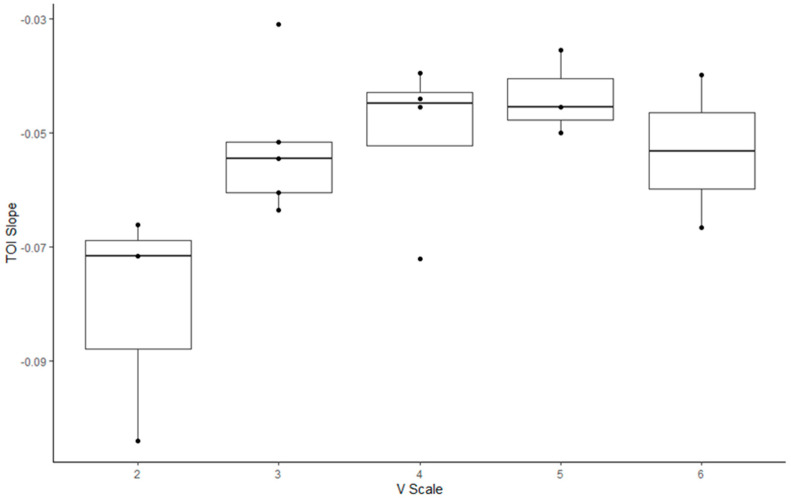
Box plot illustrating the changes in the tissue oxygenation index of rock climbers with various V-scale levels.

**Figure 5 jfmk-09-00161-f005:**
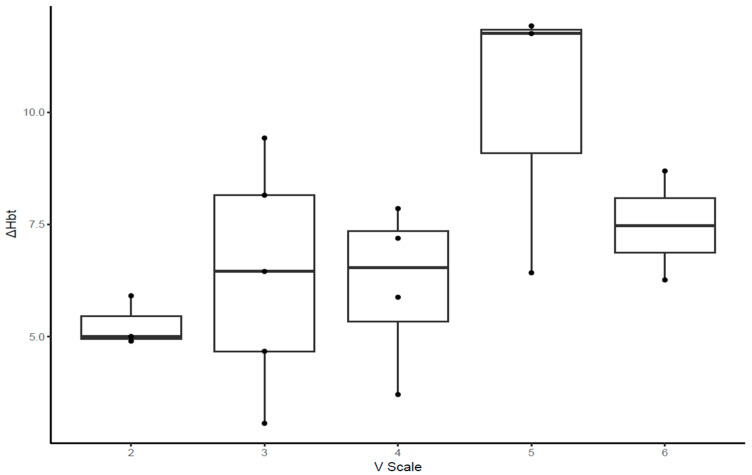
Box plot illustrating the differences between the concentrations of oxyhemoglobin and deoxyhemoglobin in rock climbers with various V-scale levels.

**Table 2 jfmk-09-00161-t002:** Participant demographics.

Characteristics	Total (*n* = 34)	Climbers (*n* = 17)	Non-Climbers (*n* = 17)	Comparison *t*-Test/Chi-Square
	Mean ± SD (Range)			*p*-Value
Age (year)	27.4 ± 3.8 (21–36)	26.4 ± 4.3 (21–35)	28.8 ± 2.8 (24–36)	*t* = 2.0, *p* = 0.061
BMI (kg/m^2^)	19.6 ± 1.8 (15.6–23.8)	19.2 ± 1.3 (17.2–22.0)	20.0 ± 2.0 (15.6–23.8)	*t* = 1.4, *p* = 0.17
V-Scale	N/A	3.8 ± 1.3 (2–6)	N/A	
Climbing experience (year)	N/A	2.4 ± 1.9 (1–8)	N/A	
Frequency of climbing (times/week)	N/A	2.3 ± 0.6 (2–4)	N/A	
	Number (%)			
Sex (male/female)	20 (58.8)/14 (41.1)	12 (70.6)/5 (29.4)	8 (47.1)/9 (52.9)	χ^2^ = 1.1, *p* = 0.29

**Table 3 jfmk-09-00161-t003:** Test–retest reliabilities of the outcome measures.

Variables	ICC(3,1)	95% CI	
		Upper	Lower
MDF slope	0.850	0.922	0.722
TOI slope	0.726	0.853	0.521
ΔTOI	0.653	0.810	0.408
Oxygen change (%)	0.749	0.866	0.556
ΔHbt	0.647	0.806	0.402

**Table 4 jfmk-09-00161-t004:** Between-group comparisons of endurance indices, * *p* < 0.05.

Variables	Mean ± SD			*t*-Test *p*-Value	Cohen’s d
	Total (*n* = 34)	Climber (*n* = 17)	Non-Climbers (*n* = 17)		
MDF Slope				0.01 *	0.93
Pre	−1.12 ± 0.48	−0.93 ± 0.46	−1.32 ± 0.43		
Post	−1.10 ± 0.43	−0.93 ± 0.39	−1.27 ± 0.42		
TOI Slope				0.02 *	0.91
Pre	−0.06 ± 0.03	−0.05 ± 0.01	−0.08 ± 0.04		
Post	−0.07 ± 0.03	−0.06 ± 0.02	−0.08 ± 0.03		
ΔTOI				0.07	0.16
Pre	0.41 ± 0.10	0.42 ± 0.08	0.40 ± 0.11		
Post	0.42 ± 0.12	0.42 ± 0.05	0.41 ± 0.16		
O_2_ Change (%)				0.07	0.68
Pre	−0.90 ± 0.08	−0.87 ± 0.10	−0.93 ± 0.05		
Post	−0.91 ± 0.07	−0.89 ± 0.07	−0.92 ± 0.05		
ΔHbt				0.01 *	0.93
Pre	5.89 ± 3.81	7.03 ± 3.59	4.76 ± 3.78		
Post	5.42 ± 2.65	6.76 ± 2.03	4.08 ± 2.55		

**Table 5 jfmk-09-00161-t005:** Correlations between the V-scale and five indices in the climbers. * *p* < 0.05.

Variables	Spearman’s Rank Correlation Rho	*p*-Value
MDF Slope	0.632	0.007 *
TOI Slope	0.533	0.028 *
ΔTOI	0.092	0.724
O_2_ Change (%)	0.195	0.454
ΔHbt	0.527	0.030 *

## Data Availability

The dataset analyzed in this study has been uploaded to Figshare and can be accessed at https://figshare.com/s/423d2043d4aa3e19ea72 (accessed on 30 August 2024).

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
