# Peer review of "Reliability and Convergent Validity of Endurance Indices Derived from Near-Infrared Spectroscopy and Electromyography during a Bilateral Hanging Task in Amateur Rock Climbers"

_jfmk, 2024, doi:10.3390/jfmk9030161_

Round 1

Reviewer 1 Report

Comments and Suggestions for Authors

Dear authors,

The manuscript is interesting with great potential. I must address some of my concerns though.

- How did you get the k constant in your formula? Can it vary across participants or is it a device parameter? Clarify in your text.

- How did you assess the asumption checks Normality and homogeneity? Clarify. 

- Set the differences between climbers and non-climbers on table 2.

- I suggest you to use 95% CI in your upper and lower limits of Bland Altman graphs. Also, put the numbers (bias, upper and lower limits) in the legend.

- Please, address in the data analysis section how did you qualitatively classify the ICC and the effect size. Insert a reference.

-  I strongly suggest you to deposit the raw data in a public repository, as Mendeley Data to improve transparency.

Author Response

Response to Reviewer 1 Comments

Thank you very much for taking the time to review this manuscript. Please find the detailed responses below and the corresponding revisions highlighted in red in the re-submitted files.

Comments 1: How did you get the k constant in your formula? Can it vary across participants or is it a device parameter? Clarify in your text.

Response 1: Thank you for pointing this out. The constant "k" represents a proportionality factor used to scale the concentrations of HbO and deoxygenated HHb. However, the value of k is not important as it was cancelled out in the formula. Although "k" cancels out in the formula, it wwas included in the formula to acknowledge the general form of such equations (Page 6 line 215-218)

Comments 2: How did you assess the asumption checks Normality and homogeneity? Clarify. 

Response 2: The information has been added in the text (page 6 line 223-225)

Comments 3: Set the differences between climbers and non-climbers on table 2

Response 3: Thanks comparisons of the two groups have been added to the table 2 (Page 6 Line 244)

Comments 4: I suggest you to use 95% CI in your upper and lower limits of Bland Altman graphs. Also, put the numbers (bias, upper and lower limits) in the legend.

Response 4: Thank you for the suggestion; the 95%CI difference is displayed in the plots now. As suggested by another reviewer, the Bland Altman was moved to supplementary materials.

Comments5: Please, address in the data analysis section how did you qualitatively classify the ICC and the effect size. Insert a reference.

Response 5: Thanks; the classification and relevant references were added to the manuscript. (Page 6, 231 – 235)

Comments 6: I strongly suggest you to deposit the raw data in a public repository, as Mendeley Data to improve transparency.

Response 6: Thank you for the suggestion. We have now uploaded the relevant raw data to Figshare. Unfortunately, one of the computers used for data capture malfunctioned, and we are currently unable to extract the NIRS data for three participants. We are working on recovering the computer, and we will upload the missing data as soon as it is retrieved. (page 12, Line 427)

Reviewer 2 Report

Comments and Suggestions for Authors

Dear Authors,

I have had the opportunity to review your article on the evaluation of the reliability and validity of muscle endurance indices derived from electromyography (EMG) and near-infrared spectroscopy (NIRS) during a bilateral suspension task in amateur climbers.

I commend the novelty of conducting studies on a sample as unique as amateur rock climbers and for applying such innovative techniques as electromyography (EMG) and near-infrared spectroscopy (NIRS). Overall, I have a very positive opinion of the manuscript and the work conducted, so I extend my congratulations. Although there are certain parts that could improve the manuscript, which I detail below, I believe that with a brief review, you would be able to understand my humble opinion and that these suggestions are easy to implement.

Abstract

I believe it is well-written, seamlessly connecting all the concepts and the rationale.

Keywords

I leave it to the authors’ discretion, but I think a good keyword to include for this study could be “amateur.” I believe it could be interesting for meta-analyses or systematic reviews.

Introduction

The introduction of the sport is very well done, presenting the key points, how the techniques are used, and why it is necessary to use both techniques. However, there are two points that I would like to see developed in the next version:

  1. In the penultimate paragraph, it is stated that “there is a gap in the literature regarding the reliability,” but when reading the introduction, I see that both EMG and NIRS have been used with this type of sample or sport, showing satisfactory results. So, what is the gap? Is it that more studies are needed to improve reliability? Is it more for looking differences in amateur people?
  2. In the last paragraph of the hypothesis, “hanging task” is mentioned. What is the particularity of this type of task? Why is it used in the study, what advantages does it offer? Is it because it may be more similar to a real context, or is it commonly used in this type of research? My point is that the authors should consider that the general reader will know about sports and physical activity, but not specifically about rock climbing, so there should be writing that can be understood in a general way. Also, if they are using an amateur sample that is unfamiliar with rock climbing, I would also mention in the hypothesis what should be observed between the two, what the difference would be.

Methods

Regarding the sample, it is stated that to be considered amateur, the criterion is “participating in rock climbing activities at least twice a week.” But what if I have been rock climbing for 30 years? Am I still considered an amateur?

It is important to consider the years of practice that the athletes have, unless it has been established that, for example, having less than 2 years of experience is sufficient to be categorized as amateur. If they have more years, I would not consider them amateur, so this is a piece of data I would include in the demographics and perhaps even conduct an analysis by skill levels, which could affect the results.

Excuse me, after observing Table 2, I see that years of experience were indeed considered. It would be good to include in the inclusion criteria up to what year was the limit to be considered amateur, and the rationale behind that established age.

Data Analysis and Results

In the Data Analysis section, I would suggest following the same order as in the Results section. It is mentioned that t-tests or Mann-Whitney tests will be performed, but the results start with Bland-Altman. It would also be necessary to include, although it may be assumed, the p-value, the software used for the analysis, and a question I have: why did the authors not calculate the effect size?

The Results section seems correct to me, but I think there are too many figures. There are a total of 10; could some be combined or eliminated? For example, perhaps not so many Bland-Altman plots are necessary.

Discussion and conclusion

Both sections seem perfectly constructed to me; congratulations. A small nuance: there is much discussion, including in the conclusion, about discerning or finding differences between climbers and non-climbers. However, I believe it is important to emphasize that your work focuses on amateur climbers. In other words, even with very little experience, you can already find differences. I think this is an important aspect that adds value to your study, if I may offer my opinion.

Author Response

Response to Reviewer 2 Comments

Thank you very much for taking the time to review this manuscript. Please find the detailed responses below and the corresponding revisions highlighted in red in the re-submitted files.

Comments 1: I leave it to the authors’ discretion, but I think a good keyword to include for this study could be “amateur.” I believe it could be interesting for meta-analyses or systematic reviews.

Response 1: Thank you for the suggestion, this keyword was added (Page 1 line 29)

Comments 2: In the penultimate paragraph, it is stated that “there is a gap in the literature regarding the reliability,” but when reading the introduction, I see that both EMG and NIRS have been used with this type of sample or sport, showing satisfactory results. So, what is the gap? Is it that more studies are needed to improve reliability? Is it more for looking differences in amateur people? 

Response 2: Thank you for the comments. While EMG and NIRS have shown satisfactory results in previous studies, There is a need for more studies that examine the reliability of these tools in a population of amateur climbers, particularly in assessing endurance, as this could reveal important insights into how muscle adaptations differ between amateurs climber and non-climber. This study aims to address that gap by focusing on the reliability of these measures in amateur climbers and exploring potential differences in muscle endurance within this group in a sport specific task. (page 2 line 75-78)

Comments 3: In the last paragraph of the hypothesis, “hanging task” is mentioned. What is the particularity of this type of task? Why is it used in the study, what advantages does it offer? Is it because it may be more similar to a real context, or is it commonly used in this type of research? My point is that the authors should consider that the general reader will know about sports and physical activity, but not specifically about rock climbing, so there should be writing that can be understood in a general way. Also, if they are using an amateur sample that is unfamiliar with rock climbing, I would also mention in the hypothesis what should be observed between the two, what the difference would be.

Response 3: Thank you for the comments. The "hanging task" is used in this study because it closely simulates the demands of rock climbing, particularly the need for sustained grip strength and forearm endurance. This task is effective for distinguishing between climbers and non-climbers, as climbers are expected to show greater endurance due to their regular engagement in activities that strengthen these muscles. (Page 2 Line 85-91)

Comments 4: It would be good to include in the inclusion criteria up to what year was the limit to be considered amateur, and the rationale behind that established age.

Response 4: Thank you for your comment. The inclusion criteria for this study are based on whether the participant is a professional climber or engages in structured training aimed at becoming a professional, rather than on age or years of participation. The rationale behind this approach is to focus on the nature of the participant's involvement in climbing, distinguishing between those who climb recreationally (amateurs) and those who are pursuing or have pursued climbing as a professional career. We have added this clarification to the manuscript (Page 3 line 100-102)

Comments 5: In the Data Analysis section, I would suggest following the same order as in the Results section. It is mentioned that t-tests or Mann-Whitney tests will be performed, but the results start with Bland-Altman. It would also be necessary to include, although it may be assumed, the p-value, the software used for the analysis, and a question I have: why did the authors not calculate the effect size?

The Results section seems correct to me, but I think there are too many figures. There are a total of 10; could some be combined or eliminated? For example, perhaps not so many Bland-Altman plots are necessary.

Response 5: Thank you for the suggestion; the order of results reporting has been revised accordingly. The software and p value threshold are included (Page 6 line 235-236). We agree that there were too many plots, so the Bland-Altman plot has been moved to the supplementary materials. Additionally, the effect size has been calculated, and the relevant descriptions have been provided.  (Page 6, line 233 – 234; table 2)

Comments 6: I believe it is important to emphasize that your work focuses on amateur climbers. In other words, even with very little experience, you can already find differences. I think this is an important aspect that adds value to your study, if I may offer my opinion.

Response 6: Thank you for your suggestion. We have incorporated this emphasis into the conclusion, highlighting that our study reveals significant differences in forearm muscle endurance even among amateur climbers with varying levels of experience.(page 12 Line 408 – 412)

Round 2

Reviewer 2 Report

Comments and Suggestions for Authors

Dear Authors,

I have read your revised version of the article and I believe you have addressed all my suggestions. Therefore, I consider the article ready for publication. Congratulations on the excellent work.